# FEW-SHOT CLASSIFICATION WITH TASK-ADAPTIVE SEMANTIC FEATURE LEARNING

## ABSTRACT

Few-shot classification aims to learn a classifier that categorizes objects of unseen classes with limited samples. One general approach is to mine as much information as possible from limited samples. This can be achieved by incorporating data aspects from multiple modals. However, existing multi-modality methods only use additional modality in support samples while adhering to a single modal in query samples. Such approach could lead to information imbalance between support and query samples, which confounds model generalization from support to query samples. Towards this problem, we propose a task-adaptive semantic feature learning mechanism to incorporate semantic features for both support and query samples. The semantic feature learner is trained episodic-wisely by regressing from the feature vectors of the support samples. Then the query samples can obtain the semantic features with this module. Such method maintains a consistent training scheme between support and query samples and enables direct model transfer from support to query datasets, which significantly improves model generalization. We develop two modality combination implementations: feature concatenation and feature fusion, based on the semantic feature learner. Extensive experiments conducted on four benchmarks demonstrate that our method outperforms state-of-the-arts, proving the effectiveness of our method.

## 1 INTRODUCTION

Deep neural network models have gained huge success in various tasks, some of them even achieved the human-level performances (He et al., 2016; Krizhevsky et al., 2012; Simonyan & Zisserman, 2014). Nevertheless, such success relies on sufficient annotated training data, which limits the applicability of the models to new concepts with very little supervision. Few shot learning has emerged as an appealing paradigm to solve this problem (Fei-Fei et al., 2006). It aims to bridge the sample-efficiency gap between machine learning and human learning in various application fields (Wang et al., 2020), such as image classification, object detection and domain adaptation (Dixit et al., 2017; Motiian et al., 2017; Snell et al., 2017).

Few-shot learning (FSL) can classify instances from unseen classes with only limited number of labeled samples. It has witnessed significant advances in few-shot classification. In FSL setting, a series of N-way K-shot few-shot tasks (or episodes) are created, where each task (or episode) contains a support set with K labeled examples per classes as supervision, and a query set with some unlabeled samples to be predicted. Meta-learning is one popular way of developing few-shot learning strategies, which has been proved effective for few-shot classification. It leverages a series of few-shot tasks generated from base dataset to learn a set of parameters of the desired classifier that can generalize well to novel tasks (Finn et al., 2017; Ren et al., 2018; Rusu et al., 2018). Another category of approach is Metric-based method. It focuses on learning a generalizable embedding model to transform all samples into a discriminative metric space, in which, samples of the same classes are clustered together, while distances between samples of the different class are widened. And a non-parametric, distance-based metric is adopted to make classification for the unlabeled samples, which simplifies the classification process in the embedding space (Gidaris & Komodakis, 2018; Li et al., 2019a; 2017; Sung et al., 2018; Vinyals et al., 2016).

Figure 1: An example of different categories under similar background(left) and similar Interference(right). When a task contains some samples from these categories, it is easy to misclassify.

Most of few-shot classification approaches are implemented based on the visual information from images. The classification result greatly depends on the visibility and cleanliness of the target in the images. It is easy to misclassify when the target is small, inconspicuous, or the target is in a similar background, or there is a similar Interference around the target, see in Figure 1. Regarding to this problem, Yue (Yue et al., 2020) propose a causal intervention mechanism to eliminate the effect of similar background. Hou (Hou et al., 2019) proposes a cross attention network to locate the most relevant regions in the pair of labeled and unlabeled samples, which can help to highlight the target object, so as to alleviate the influence of similar Interference. Xing (Xing et al., 2019) incorporates semantic features with the features for support set samples to help distinguishing samples of different targets with similar appearance, which can modify the position of class centers. However, this method is incapable of adjusting the feature embeddings of samples from query set. Previous works have demonstrated that leveraging multiple modalities in few-shot image classification have great potential in performance improvement (Xing et al., 2019; Schwartz et al., 2019; Li et al., 2020; Chen et al., 2019).

However, what is missing from the current multi-modal approaches is that the semantic features is only available in the support set but not in query set. Consequently, separating visually similar objects of different classes in the query set, which are in close vicinities of each other in the visual feature space, remains a fairly difficult task. To solve this problem, we propose a task-adaptive semantic feature learning mechanism, to predict semantic features for the query samples which is helpful in distinguishing samples with similar background and interference. The semantic features are the word embeddings of the text labels, which are predicted by the Glove word embedding method. The task adaptive semantic feature learner is trained within each task with the support samples as the supervision. Then, the trained semantic feature learner is used to predict semantic features for the query samples, which can guide the model to pay more attention to the object of the corresponding class. The semantic feature learner is customized to respective task such that the semantic features are most discriminative for a given task. The contributions of this work are three folds. (1) We propose a task-adaptive semantic feature learner to predict semantic features for the query samples, which improves the discrimination of the query features. (2) The visual and semantic features are learned separately without collapse into a common feature space. which maintains specific information of different modalities. (3) we construct some hard tasks with samples having similar background and interference, to prove the efficacy of TasNet in generating discriminative features from hard samples. (4) We develop two modality combination implementations, feature concatenation and feature fusion (feature weighted sum), both results show the effectiveness of the semantic feature leaner.

## 2 RELATED WORK

### 2.1 FEW SHOT LEARNING

Few-shot classification methods exhibit great diversity. These methods can be roughly divided into two main groups: metric-based and gradient-based approaches. Metric-based approaches aim at learning discriminative representations that minimize intra-class distances while maximizing the inter-class distances with a non-parametric metric. Representative works include Siamese Network (Koch et al., 2015), Matching Network (Vinyals et al., 2016) and Prototypical Network (Snell et al., 2017). On this basis, Relation network (Sung et al., 2018) proposed a learnable deep distance metric, to construct the relation of samples within a task. The unlabeled samples are classified according to the metric scores. Gradient based methods focus on training models that can generalize well to new tasks with only a few fine-tuning updates. MTL (Sun et al., 2019) approach leverages transfer

learning and benefits from referencing neuron knowledge in pre-trained deep nets. MAML (Finn et al., 2017) aims to learn a good parameter initialization that enables the model easy to fine-tune.

## 2.2 MULTI-MODALITY FEW SHOT LEARNING

Recently, some researches concerning multi-modality have been proposed, such as Adaptive Modality Mixture Mechanism (AM3) (Xing et al., 2019), Multiple-Semantics (Schwartz et al., 2019), task-relevant additive margin loss (TRAML) (Li et al., 2020) and Semantic Feature Augmentation (Chen et al., 2019), to leverage extra information of the samples to improve the classification performance. In (Xing et al., 2019), Xing et al. observed that some visually-similar images of different classes are distant from each other in the semantic feature space. Thus, they proposed an adaptive feature fusion mechanism to combine features of the visual and semantic modalities to refine the category prototypes of the support set. Similarly, E. Schwartz et al. (Schwartz et al., 2019) proposed a improved version of AM3 by incorporating multiple and richer semantics information (category labels, attributes, and natural language descriptions) to the image visual features, which could result in further performance improvements. This idea came from mimicking the learning process of a human baby. TRAML (Li et al., 2020) proposed an adaptive margin principle to improve the generalization ability of metric-based meta-learning approaches. The adaptive margins are defined as the semantic similarity between different categories. TriNet (Chen et al., 2019) solves the few shot classification problem via feature augmentation in the semantic feature space. It first embedded images into a latent semantic space. And then, it augmented semantic features with semantic gaussian and semantic neighborhood approaches. After which, the semantic features are transformed back to the image visual features space, which enriches useful information for the limited samples. A common limitation of these methods is that while additional modalities are used in support samples, there is still a single modality used in the query samples. Our method solves this problem by incorporating visual and semantic features in both support and query samples.

## 3 METHODOLOGY

We incorporate data of multiple modalities as an attempt to extract additional semantic information from samples . The key challenge, however, is that labels of the query set are obstructed from the classifier. Instead of abandoning the additional modality all together in classifying samples in the query set, we choose to re-construct the missing modality from available information. This requires a translator that predicts values of the missing modality. In the realm of image classification, we train a task-specific image-to-semantic translator, termed the semantic feature learner, from the support set, which is then applied to the query set to generate the semantic features for each image. Since both the visual and the semantic modalities are now present in the query set, we combine these two features via concatenation or fusion operation to produce final feature representations for each sample.

In our method, the semantic feature learner is trained separately from the visual feature learner. The key idea of isolating the two training processes is that while visual features might be shared across all categories and many tasks, semantic features, as a supplementing feature set, is used to help distinguish instances of visually similar categories within the current task. Hence the visual feature learner is trained globally and the semantics feature learner is trained locally per each task. Depending on the task setup, the semantic feature learner might focus on different features in images. For example, the semantic feature learner might put different emphases in egg images to help separate eggs from pingpong balls (in one specific task), then from separating eggs from carrots (in another specific task). By limiting the training process of the semantic feature learner to each task, our method allows greater flexibilities adapting to each task and increases the efficacy of using the semantic features to further distinguish visually similar objects in a task.

Overall, our method trains a global visual feature learner ($f_\varphi$) across all tasks during the meta-training phase, while trains a task-specific semantic feature learner ($g_\psi$) within each task separately. In support set, we compute the visual-feature class centers (visual prototypes) and combines them with label-derived semantic features into final prototype feature vectors $\boldsymbol{p}$ (class centers). In query set, we generate both the visual features and the semantic features from $f_\varphi$ and $g_\psi$, respectively, which are then combined into a final feature representation $\boldsymbol{q}$ of the query sample. Finally, the

classification of query samples is implemented based on the distance to prototypes associated to each class.

## 3.1 PROBLEM DEFINITION

In Few Shot Learning setting, There are three datasets: $\mathcal{D}_b$, $\mathcal{D}_v$ and $\mathcal{D}_n$ denoting train (base), validation and test (novel) set respectively, with categories disjoint from each other, i.e. $\mathcal{D}_b \cap \mathcal{D}_v = \emptyset$, which is different from traditional supervised learning. The goal of FSL is to classify samples from unseen classes with few supervised samples. In general, the few shot learning approaches are based on episodic training paradigm. The model is trained across a number of episodes $\mathcal{T}$ sampled from $\mathcal{D}_b$. A $N$-way $K_S$-shot task (or episode) is denoted as $\mathcal{T}_i = \{\mathcal{S}; \mathcal{Q}\}$, containing $N$ categories. Where $\mathcal{S}$ denotes the support set consisting of $K_S$ samples per category, i.e. $\mathcal{S} = (x_i, y_i)_{i=1}^{N \times K_S}$; and $\mathcal{Q}$ denotes the query set consisting of $K_Q$ samples per category, i.e. $\mathcal{Q} = (x_i, y_i)_{i=1}^{N \times K_Q}$. Finally, the performance of the model is evaluated on multiple tasks sampled from the novel dataset $\mathcal{D}_n$.

## 3.2 ARCHITECTURE OF THE PROPOSED MODEL

Figure 2 illustrates the overall framework of the proposed method. It consists of a pre-training stage, a meta-training stage and a meta-testing stage. Firstly, we pre-train a feature extractor $E_\theta$ with $C(\cdot|\mathcal{D}_b)$ as the classifier from scratch by minimizing a standard cross entropy loss with the base dataset $\mathcal{D}_b$ as the training data. After pre-training stage, the parameters of the feature extractor are frozen, which is indicated with light blue color. In the meta-training stage, images are first fed into the feature extractor $E_\theta$ to produce feature vectors. Then, the feature vectors are used as inputs for the visual feature learner $f_\varphi$ and the semantic feature learner $g_\psi$. Meanwhile, the support images and the corresponding class text labels $(I_S, W_S)$ are feed into $F(\cdot|\mathcal{S})$ module to train the task-specific semantic feature leaner. The trained semantic feature learner $g_\psi$ is represented in light yellow color, denoting the parameters are frozen. Both the final prototypes and the final feature representations of the query samples are generated by **F C/F** module. The key difference between the support feature representations and the query feature representations is that: in the support set, semantic features are generated from the class text labels, while in the query set, semantic features are predicted with $g_\psi$ from image feature vectors. Finally, the visual feature learner $f_\varphi$ is optimized via minimizing the cross-entropy loss between the ground truth labels and the predicted probability vectors. The meta-testing stage is similar with the meta-training stage, except that the parameters of the visual feature learner $f_\varphi$ can be no longer updated, which is represented in light green color.

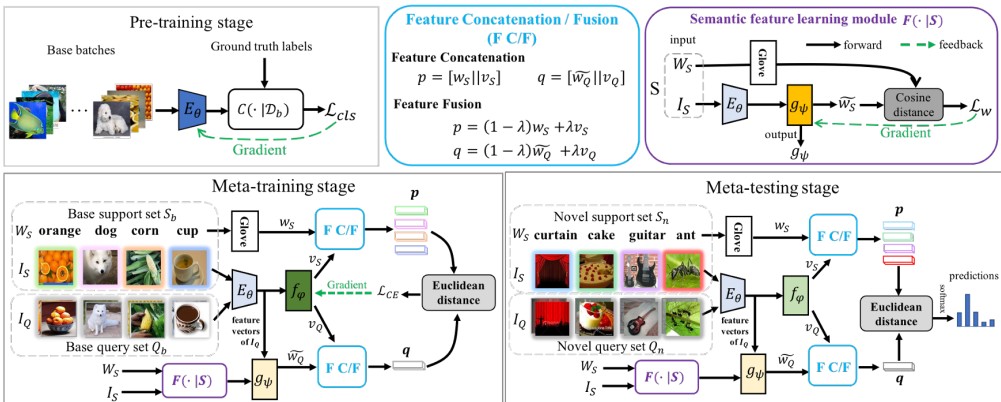

Figure 2: The overall architecture of the few shot classification method with task-adaptive semantic feature learning mechanism.

## 3.3 TASK-ADAPTIVE SEMANTIC FEATURE LEARNING

The feature vectors of the support images and the ground truth semantic feature embeddings (it is predicted by Glove word embedding of the class text labels) are paired up as supervision to train the semantic feature learner $g_\psi$. The loss function is formulated as the dissimilarity between the

predicted semantic feature embeddings and the ground-truth semantic feature embeddings. See in Eq. 1.

$$\mathcal{L}_{dissim}\left(\psi\right) = \sum_{c=1}^{N}\sum_{i=1}^{K_S}\left(1 - \frac{\widetilde{w_{c_i}} \cdot w_c^T}{||\widetilde{w_{c_i}}|| \cdot ||w_c||}\right) \tag{1}$$

where $\widetilde{w_{c_i}}$ denotes the predicted semantic feature embedding of the $i$-th sample belonging to class $c$, and $w_c$ denotes the ground truth semantic feature embedding of class $c$. We train the semantic feature learner within each task by minimizing $\mathcal{L}_{dissim}$ to produce task-specific parameters.

### 3.4 FEATURE GENERATION

#### 3.4.1 PROTOTYPES GENERATION OF THE SUPPORT SAMPLES

Firstly, we extract feature vectors for both support and query samples, as formulated in Eq. 2.

$$a_i = E_\theta\left(x_i\right), \quad i = 1, \ldots, N(K_S + K_Q) \tag{2}$$

where $x_i$ denotes an image sample, and $a_i$ is the feature vector extracted from this image. Then we transform the feature vectors to the visual feature space via the visual feature learner $f_\varphi$: $v = f_\varphi\left(a\right)$. And we compute the visual prototypes with Eq. 3

$$v_c = \frac{1}{K_S} \sum_{(x_j,\, y_j) \in \mathcal{S}_c} f_\varphi\left(a_j\right) \quad c = 1, ..., N \tag{3}$$

where $\mathcal{S}_c$ denotes a subset of $\mathcal{S}$, which contains samples of class $c$.

Finally, we combine the visual prototypes with the ground truth semantic feature embeddings via concatenation or fusion (weighted sum) operation to obtain the final prototypes of the support set. It is formulated as Eq. 4 and Eq. 5, respectively.

$$p_c = \mathcal{C}\left(v_c, w_c\right), \quad c = 1, \ldots, N \tag{4}$$

$$p_c = \lambda v_c + (1 - \lambda)w_c, \quad c = 1, \ldots, N \tag{5}$$

where $\mathcal{C}(\cdot, \cdot)$ denotes the concatenation operation of the two vectors, $w_c$ is the ground truth semantic feature embedding of class $c$.

#### 3.4.2 FEATURE GENERATION OF QUERY SAMPLES

The feature representations of the query samples are produced in a similar way as support samples, except that the semantic features of the query samples are learned from images. The final query feature representations are generated via fusion approach formulated as Eq. 6 or concatenation approach formulated as Eq. 7

$$q_i = \mathcal{C}\left(f_\varphi(a_i),\, g_\psi\left(a_i\right)\right), \quad i = 1, \ldots, NK_Q \tag{6}$$

$$q_i = \lambda f_\varphi(a_i) + (1 - \lambda)\, g_\psi\left(a_i\right), \quad i = 1, \ldots, NK_Q \tag{7}$$

where $a_i$ is the feature vector of $i$-th query image sample.

---

**Algorithm 1:** Task adaptive semantic feature learning for few shot classification

---

    **Input:** Task set $\mathcal{T} = \{\, \mathcal{T}_i = \{\mathcal{S}; \mathcal{Q}\} \,|\, i = 1, ..., N_T \}$

    **Output:** visual feature learner $f_\varphi$ and semantic feature learner $g_\psi$

1  Randomly initialize $\varphi$ and $\psi$;

2  **for** *t in* $\{\mathcal{T}_1, \ldots, \mathcal{T}_{N_T}\}$ **do**

3     Compute prototypes $\boldsymbol{p}$ for the support samples:

4     **for** *c in 1, ..., N* **do**

5         Extract feature vectors of images from $\mathcal{S}$ by Eq. 2;

6         Compute visual prototypes $v_c^s$ by Eq. 3;

7         Combine the semantic and visual features by Eq. 4 or Eq. 5;

8         Compute semantic features of $\mathcal{S}$ with $g_\psi$;

9     **end**

10    Optimize parameters of $g_\psi$ with samples from $\mathcal{S}$ by **Algorithm 2**;

11    Compute features $\boldsymbol{q}$ of query samples:

12    **for** *j in* $\{1, \ldots, NK_Q\}$ **do**

13       Extract feature vectors $a_j^q = E_\theta\left(x_j^q\right)$ of samples from $\mathcal{Q}$;

14       Compute visual features $v_j^q = f_\varphi(a_j^q)$;

15       Predict semantic features with $g_\psi(a_j^q)$;

16       Compute final feature representations of query samples by Eq .6 or Eq .7;

17    **end**

18    Compute the cross entropy loss $\mathcal{L}_{CE}$ using Eq. 9;

19    Update $\varphi$ by gradient descent $\nabla \mathcal{L}_{CE}$.

20 **end**

---

**Algorithm 2:** Task-specific semantic feature learner optimization

---

    **Input:** $W = \left\{ (w_c; w'_{c_i}) \,|\, c = 1, \ldots, N,\ i = 1, \ldots K_S \right\}$. ($w_c$ is the ground truth semantic features and $w'_{c_i} = g_\psi\left(a_{c_i}\right)$ is the predicted semantic features.)

    **Output:** semantic feature learner $g_\psi$

1  **while** *not convergence* **do**

2     Evaluate $\mathcal{L}_{dissim}\left(\psi\right)$ by Eq. 1;

3     Update $\psi$ by gradient descent $\nabla \mathcal{L}_{dissim}$;

4 **end**

---

### 3.4.3   CLASSIFICATION LOSS OF THE VISUAL FEATURE LEARNER

The negative Euclidean distance between the feature representation $q_i$ of *i*-th query and the prototype $p_c$ of the *c*-th support sample is employed as the predicted scores of the query samples, and the distance is transformed into a probability via a Softmax operation, see Eq. 8.

$$
\begin{aligned}
d\left(q_i,\ p_c\right) &= \|\, q_i - p_c \,\|_2\,, \quad i = 1, \ldots, NK_Q \\
p\left(\hat{y}_i = c | q_i, \varphi\right) &= \frac{e^{-d(q_i,\ p_c)}}{\sum_{c=1}^{N} e^{-d(q_i,\ p_c)}}\,, \quad c = 1, \ldots, N
\end{aligned}
\tag{8}
$$

where $\hat{y}_i$ denotes the predicted label for *i*-th sample in the query set. Then, the loss function is formulated as the negative loglikelihood of the predicted probability scored of each query sample.

$$
\mathcal{L}_{CE}\left(\varphi\right) = - \sum_{i=1, x_i \in \mathcal{Q}}^{K_Q} y_i \log\left(p\left(\hat{y}_i = y_i | x_i, \varphi\right)\right)
\tag{9}
$$

We compute the cross-entropy loss between the predicted scores of the query samples and the ground truth labels for all training episodes constructed from the base dataset, see Eq. 9. Parameters of the visual feature learner is optimized by minimizing $\mathcal{L}_{CE}$. We summarize the whole process in Algorithm 1 and Algorithm 2.

Table 1: The few-shot classification accuracy results on MiniImageNet and CIFAR-FS datasets. They present the mean accuracy on 600 novel episodes with a 95% confidence interval. The numbered superscripts denote the architecture of the feature extractor: [1]ResNet12 (Franceschi et al., 2018) , [2]ResNet18 (Li et al., 2019b), [3]ResNet25, [4]WRN-28-10 (Zagoruyko & Komodakis, 2016)."−" indicates that the model has not run experiments on this dataset.

| Model | miniImageNet 5-way | | CIFAR-FS 5-way | |
|---|---|---|---|---|
| | 1-shot | 5-shot | 1-shot | 5-shot |
| MatchingNet[1](Vinyals et al., 2016) | 63.08 ± 0.80 | 75.99 ± 0.60 | − | − |
| ProtoNet[1](Snell et al., 2017) | 60.37 ± 0.83 | 78.02 ± 0.57 | 72.20 ± 0.70 | 83.50 ± 0.50 |
| TADAM[1](Oreshkin et al., 2018) | 58.50 ± 0.30 | 76.70 ± 0.30 | − | − |
| MTL[1](Sun et al., 2019) | 61.20 ± 1.80 | 75.50 ± 0.80 | − | − |
| MetaOptNet[1] (Lee et al., 2019) | 62.64 ± 0.61 | 78.63 ± 0.46 | 72.0 ± 0.70 | 84.20 ± 0.50 |
| FEAT[1] (Ye et al., 2020) | 66.78 ± 0.20 | 82.05 ± 0.14 | − | − |
| AM3-TADAM[1] (Xing et al., 2019) | 65.30 ± 0.49 | 78.10 ± 0.36 | − | − |
| DeepEMD[1] (Zhang et al., 2020) | 65.91 ± 0.82 | 82.41 ± 0.56 | − | − |
| E3BM[3] (Liu et al., 2020b) | 64.30 ± 0.90 | 81.0 ± 0.90 | − | − |
| ConstellationNet (Xu et al., 2020) | 64.89 ± 0.23 | 79.95 ±0.17 | 75.40 ± 0.20 | 86.80 ± 0.20 |
| PSST[4] (Chen et al., 2021) | 64.16 ± 0.44 | 80.64 ± 0.32 | 77.02 ± 0.38 | **88.45 ± 0.35** |
| Neg-Cosine[2] (Liu et al., 2020a) | 63.85 ± 0.81 | 81.57 ± 0.56 | − | − |
| MABAS[1] (Kim et al., 2020) | 65.08 ± 0.86 | 82.70 ± 0.54 | 73.51 ± 0.92 | 85.49 ± 0.68 |
| *MetaOptNet + ArL*[1] (Zhang et al., 2021) | 65.21 ± 0.58 | 80.41 ± 0.49 | − | − |
| TasNet[1] (concatenation) | **78.68 ± 0.59** | **85.89 ± 0.47** | **80.80 ± 0.70** | 84.86 ± 0.60 |
| TasNet[1] (fusion) | 78.24 ± 0.64 | 85.82 ± 0.49 | 80.14 ± 0.72 | 84.46 ± 0.58 |
| TADAM +Tasnet[1] | 78.57 ± 0.56 | 85.57 ± 0.41 | 80.47 ± 0.67 | 84.57 ± 0.59 |
| MetaOptNet +Tasnet[1] | 78.36 ± 0.61 | 85.78 ± 0.52 | 80.76 ± 0.73 | 84.69 ± 0.63 |

## 4 EXPERIMENTS

We conduct a comparison of our method to state-of-the-art methods in terms of few-shot classification accuracy on four benchmarks, including MiniImageNet (Vinyals et al., 2016), CUB-200-2011 (Wah et al., 2011), CIFAR-FS (Bertinetto et al., 2018) and FC100 (Oreshkin et al., 2018).

### 4.1 ARCHITECTIRES

For the feature extractor network, we use a ResNet-12 consisting of 4 residual blocks with Dropblock as a regularizer and 640-dimensional output features, the details follow the one proposed in TADAM (Oreshkin et al., 2018). For the visual feature learner and the semantic feature learner, we use a MLP with one layer, outputting 300-dimentional features.

### 4.2 RESULTS AND ANALYSIS

Table 1 and Table 2 summarize the results of the few-shot classification tasks on MiniImageNet, CIFAR-FS, FC100, and CUB-200-2011, respectively. It is worth noting that our TasNet achieves the state-of-the-art performance for both 1-shot and 5-shot settings on almost all four benchmarks.

### 4.3 ABLATION STUDY

To further validate the effectiveness of our method, we conduct a series of ablation studies on mini-ImageNet and CIFAR-FS datasets. We first show some experimental proofs of the task-adaptive semantic feature learner. Then we show some other results on the hyperparameters.

#### 4.3.1 EFFECTIVENESS OF THE SEMANTIC FEATURE LEARNER

The task-adaptive semantic feature learner is designed to alleviate the negative effects of similar background or similar interference, and provide another kind of modality to enrich the information of query samples, which is helpful in producing discriminative feature embeddings. It can also alleviate the few supervision problem to a certain extent. We test the model with and without semantic feature learner. The quantitative and qualitative results are shown in Figure 3(a) and Figure 4.

Table 2: The few-shot classification accuracy results on FC100 and CUB-200-2011 datasets. They present the mean accuracy on 600 novel episodes with a 95% confidence interval. The numbered superscripts have the same meaning as that in Table 1

| Model | FC100 5-way | | CUB 5-way | |
|---|---|---|---|---|
| | 1-shot | 5-shot | 1-shot | 5-shot |
| MatchingNet[1] (Vinyals et al., 2016) | – | – | 72.40 ± n/a | 83.60 ± n/a |
| ProtoNet[1] (Snell et al., 2017) | 41.50 ± 0.70 | 57.10 ± 0.70 | 71.90 ± n/a | 87.40 ± n/a |
| TADAM[1] (Oreshkin et al., 2018) | 40.10 ± 0.40 | 56.10 ± 0.40 | – | – |
| MTL[1] (Sun et al., 2019) | 45.10 ± 1.80 | 57.60 ± 0.90 | – | – |
| MetaOptNet[1] (Lee et al., 2019) | 41.10 ± 0.60 | 55.50 ± 0.60 | – | – |
| FEAT[1] (Ye et al., 2020) | – | – | 68.60 ± n/a | 83.0 ± n/a |
| DeepEMD[1] (Zhang et al., 2020) | 46.50 ± 0.80 | 63.20 ± 0.70 | 79.30 ± 0.30 | **89.80 ± 0.5** |
| E3BM[3] (Liu et al., 2020b) | – | – | 45.0 ± 1.30 | 60.50 ± 0.4 |
| ConstellationNet[1] (Xu et al., 2020) | 43.80 ± 0.20 | 59.70 ± 0.20 | – | – |
| Neg-Cosine[2] (Liu et al., 2020a) | – | – | 72.66 ± 0.85 | 89.40 ± 0.43 |
| MABAS[1] (Kim et al., 2020) | 42.31 ± 0.75 | 57.56 ± 0.78 | – | – |
| *SoSN + ArL*[1] (Zhang et al., 2021) | – | – | 50.62 ± n/a | 65.87 ± n/a |
| TasNet[1] (concatenation) | **61.60 ± 0.83** | **68.51 ± 0.69** | **79.89 ± 0.74** | 85.76 ± 0.63 |
| TasNet[1] (fusion) | 60.98 ± 0.89 | 67.66 ± 0.84 | 79.17± 0.67 | 84.99 ± 0.66 |
| TADAM +Tasnet[1] | 61.79 ± 0.78 | 68.35 ± 0.65 | 79.72 ± 0.59 | 86.12 ± 0.68 |
| MetaOptNet +Tasnet[1] | 61.24 ± 0.91 | 67.98 ± 0.73 | 79.32 ± 0.71 | 84.75 ± 0.72 |

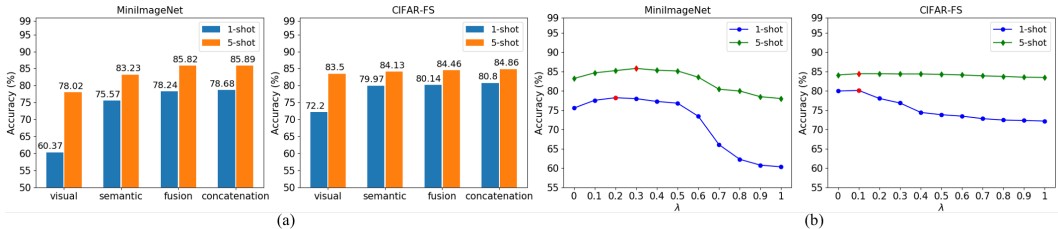

Figure 3: (a) Quantitative comparison of the effect of different features in terms of the mean accuracy on 600 novel episodes on 5-way 1-shot and 5-way 5-shot cases for both MiniImageNet and CIFAR-FS datasets, where *visual* indicates the visual features learned by the visual feature learner $f_\varphi$, *semantic* indicates the semantic features learned by the semantic feature learner $g_\psi$, *fusion* denotes the weighted sum of the visual and semantic feature embeddings, *concatenation* denotes the concatenated features of the visual and semantic feature embeddings. (b) Hyperparameter analysis for the fusion approach for the semantic and visual features. We study the effectiveness of changing the value of fusion weight $\lambda$ on the mean accuracy of 600 novel episodes in terms of 1-shot and 5-shot cases. Where the *red* points denote the optimal hyperparameters for the respective cases.

As shown in Figure 3 (a), the classification accuracy has achieved a reasonable value in the case of only semantic features. In the 1-shot case, the contribution of the semantic feature is especially significant, which proves the effectiveness of the task-adaptive semantic feature learner. We can also observe that, the combination of the semantic and visual features resulting in further performance improvement, which demonstrates that the semantic features and the visual features can complement each other to improve the classification problem together. Moreover, performance of the concatenation approach of features is ahead of the fusion approach.

### 4.3.2 T-SNE VISUALIZATION

To qualitatively validate the effectiveness of our method, we further provide some t-SNE visualization results in Figure 4. We sample one episode in the test split of miniImageNet and CIFAR-FS datasets under the 5-way 1-shot and 5-way 5-shot settings. And then, we obtain the embeddings of all images using the visual feature learner, the semantic feature learner, and our TasNet, respectively. Across four subfigures, samples with the same color belong to the same class. Evident in these figures, the overlap between different classes of embeddings generated by TasNet is much less than

that with only visual features, which demonstrates the effectiveness of the task-adaptive semantic feature learner in helping learning feature representations with well separated margin between novel classes. From the last two columns of Figure 4 (a)(b), we can further observe that compared with the visualization of the fused feature embeddings, the discriminative performance of the concatenated feature embeddings is a little superior than the former, which is in consistent with the quantitative result in Figure 3(a) .

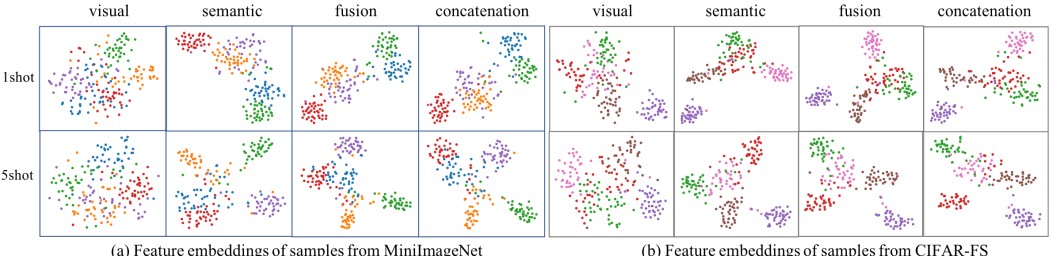

(a) Feature embeddings of samples from MiniImageNet      (b) Feature embeddings of samples from CIFAR-FS

Figure 4: t-SNE visualization of the feature embeddings generated by TasNet in 5-way 1-shot and 5-way 5-shot cases with 40 query samples per class from novel classes on MiniImageNet and CIFAR-FS datasets. the x-axis notations have the same meaning as that in Figure 3(a)

In Figure 3(b), we vary the value of fusion weight $\lambda$ from 0 to 1 to observe the performance change. This hyperparameter controls the contribution of the semantic and visual features to the final fused features. As shown in Figure 3(b), we can observe that, the optimal $\lambda$ for MiniImageNet in 1-shot and 5-shot cases are 0.2 and 0.3, respectively, while for the CIFAR-FS dataset, the optimal $\lambda$ reaches at 0.1 for both 1-shot and 5-shot cases. It demonstrates that, in order to improve the final classification performance, we should weigh more on the semantic features, while information from the visual side should be relatively low. Since too much attention paid to the visual feature will weak the advantage brought by the semantic features. It is in consistent with the results of the ablation studies in the former sections. This indicates that the selection of $\lambda$ plays a key role for best complementing the features information of different modalities.

From the quantitative and qualitative results, we can conclude that the task-adaptive semantic feature learning mechanism is the main reason for the performance boosting.

## 5    CONCLUSION

In this paper, we propose TasNet, a task adaptive semantic feature learning mechanism to incorporate semantic features for the query samples. Since the feature extractor has been initialized after the pre-training stage, we freeze the parameters of the feature extractor in the meta-training and meta-testing stages. Therefore, only the semantic feature learner and the visual feature learner should be optimized, which is less time consuming. Our model trains the global classification loss and the task-adaptive semantic feature dissimilarity loss separately, which can avoid collapsing different modalities into a common feature space, so as to preserve the structural heterogeneity of different modalities. Then, we propose two features combination approaches: feature concatenation and feature fusion, to further improve the performance. Experiments show that our method outperforms the state-of-the-arts on four benchmarks. In the future work, we expect to explore semantic-guided attention mechanism to pay more attention to the targets in the images, so as to better capture class-specific information from the images, which can further alleviate the influence of noise and irrelevant information.

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

## A   APPENDIX

### A.1   SETTINGS

We train the feature extractor $E_\theta$ with the SGD optimizer using an initial learning rate of $10^{-1}$. Adam (Kingma & Ba, 2014) is used to optimize the parameters of the visual and semantic feature learners with an initial learning rate of $10^{-3}$. During few-shot learning stage, we train $f_\varphi$ and $g_\psi$ by adopting the episodic training procedure with N-way K-shot training tasks as described in (Snell et al., 2017). The model is trained for 100 epochs, with each epoch consisting of 600 randomly sampled episodes for both 1-shot and 5-shot cases on MiniImageNet, CUB-200-2011, CIFAR-FS and FC100 datasets, and is evaluated by averaging metrics over 600 randomly generated episodes from $\mathcal{D}_n$, with each episode having 15 randomly sampled query samples. Parameters of the model except the backbone layers are trained on 5-way, 15 query samples for both 1-shot and 5-shot cases per episode.

### A.2   EXPERIMENTS ON HARD TASKS

To further validate the effectiveness of TasNet in discriminating different targets from similar background and interference, we conduct an additional experiment on hard tasks, which are constructed of samples with similar background and interference.

Firstly, we have constructed a sub dataset (we name it hard dataset) from the test split of miniImagenet dataset manually. It contains 8 classes, including lion, golden retriever, goose, coral reef, vase, spider web, ant, house finch. Among samples from all these categories, some of samples from two or more categories have similar background or similar interference, for example, vase and ant categories, flowers are the common interference in the images from these two classes, and for the bird and spider web categories, trees and grassland are common similar backgrounds in the images from these two classes. Then we construct 20 tasks from this hard dataset, and compare our TasNet with ProtoNet and AM3 methods on this dataset. AM3 is a method that combine semantic features for the support samples, while only visual features are extracted from the query samples. The mean accuracy is show in table 3.

Table 3: The few-shot classification accuracy results on hard datasets sampled from miniImageNet. They present the mean accuracy on 50 novel episodes with a 95% confidence interval.

| Model | Hard dataset 5-way | |
| --- | --- | --- |
| | 1-shot | 5-shot |
| ProtoNet (Vinyals et al., 2016) | 44.93 ± 0.71 | 70.67 ± 0.58 |
| AM3 (Snell et al., 2017) | 48.11 ± 0.69 | 71.24 ± 0.61 |
| TasNet | 50.97 ± 0.76 | 72.02 ± 0.69 |

In addition to the statistical results, we show the classification results of one hard task in Figure 5, which is carried on 5-way 1-shot case. Five categories of ant, house finch, spider web, vase, coral reef, are included in this task. Each class has 1 support samples and 15 query samples, that is the number of test samples is 5×15=75.

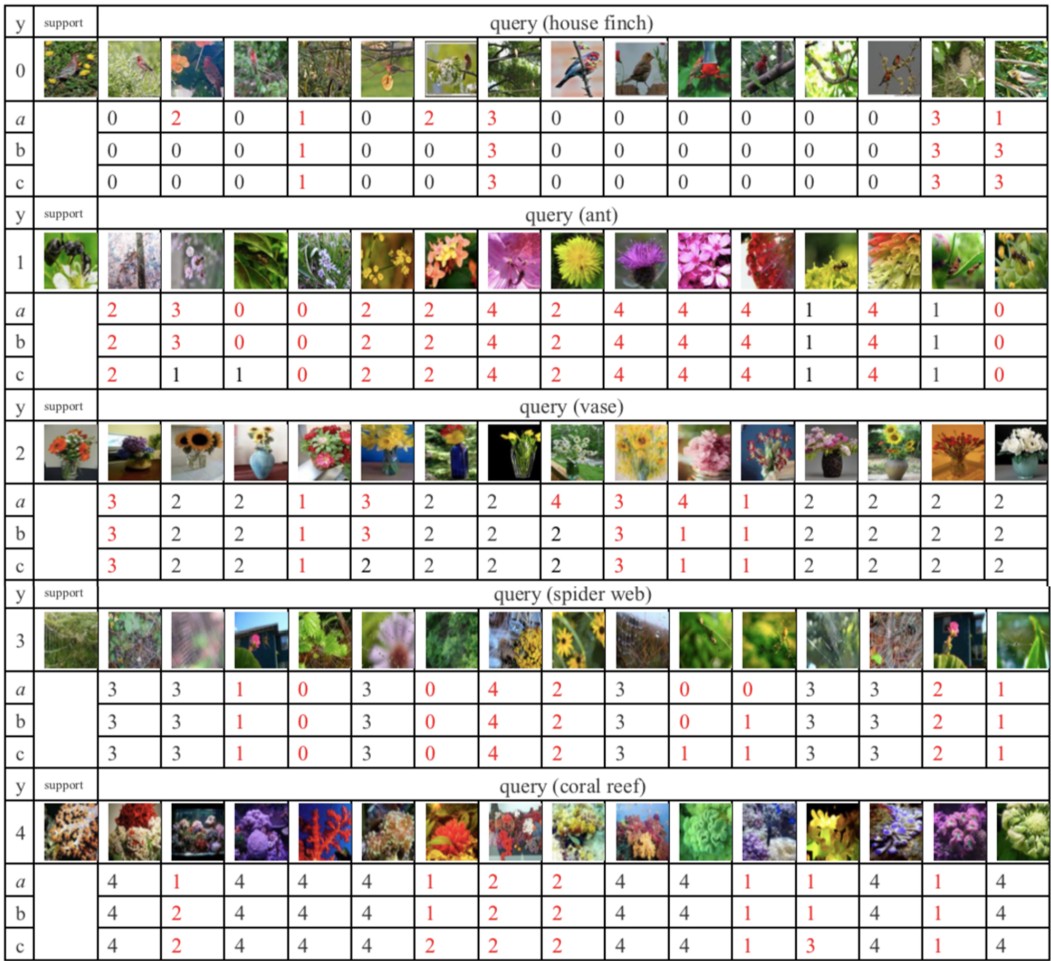

Figure 5: Detailed classification result of one hard task. where a denotes ProtoNet, b denotes AM3, c denotes TasNet. The number ID 0-4 below y denotes the class labels. The red number means the sample is miss classified.

From Figure 5, we can see that there are 33, 36 and 39 samples correctly classified with the ProtoNet, AM3 and our TasNet methods.

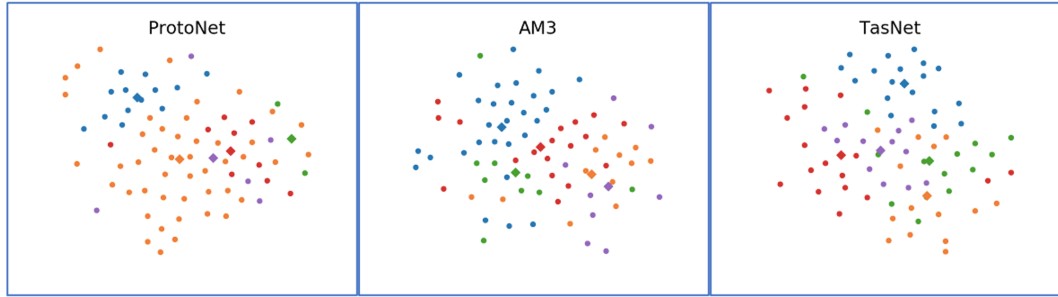

Figure 6: TSNE visualization of the query samples from one hard task.

### A.3 ANALYSIS OF THE ABLATION STUDY RESULTS

From the quantitative and qualitative results in section 4.3.1 and section 4.3.2, we can conclude that the task-adaptive semantic feature learning mechanism is the main reason for the performance

boosting. To make the ablation study results more obvious, we put the accuracy in terms of four kinds of different features in table 4.

Table 4: Ablation study results

| Features | | | | miniImageNet 5-way | | CIFAR-FS 5-way | |
|---|---|---|---|---|---|---|---|
| visual | semantic | fused | concatenated | 1-shot | 5-shot | 1-shot | 5-shot |
| ✓ | | | | 60.37 | 78.02 | 72.2 | 83.5 |
| | ✓ | | | 75.57 | 83.23 | 79.97 | 84.13 |
| ✓ | ✓ | ✓ | | 78.24 | 85.82 | 80.14 | 84.46 |
| ✓ | ✓ | | ✓ | 78.68 | 85.89 | 80.8 | 84.86 |

The accuracy computed with the semantic features is shown in the second row of table 4 (in red color). When only the semantic features are used in the model, it has achieved a significant accuracy, which demonstrates that the semantic features for the query samples can certainly work.

## A.4 TRAINING THE SEMANTIC AND VISUAL FEATURE LEARNER JOINTLY

In order to prove that training the visual feature learner across all tasks and training the semantic feature learner within each task is an effective approach, we have made some experiments to jointly train the visual and semantic features learners. The detailed implementation of the jointly training method is described as follow.

We constructed a cross entropy loss to optimize the visual feature learner, and a word embedding fitting loss, i.e. the cosine dissimilarity loss to optimize the semantic feature learner.

The loss $\mathcal{L}_{CE}$ for training visual feature learner is defined as a cross-entropy classification loss averaged across all query-support pairs. It compares the Euclidean distance between the query features and the class mean of the support features to get the predicted score.

$$d\left(q_i,\ p_c\right) = ||\ q_i - p_c\ ||_2\ , \quad i = 1,\ldots, NK_Q \tag{10}$$

$$p\left(\hat{y}_i = c | q_i, \varphi\right) = \frac{e^{-d(q_i,\ p_c)}}{\sum_{k=1}^{N} e^{-d(q_i,\ p_k)}}\ , \quad c = 1, \ldots, N \tag{11}$$

where $\hat{y}_i$ denotes the predicted label for **i**-th sample in the query set, and $y_i$ is the ground truth label of sample $q_i$. Then, the loss function is formulated as the negative loglikelihood of the predicted probability scored of each query sample.

$$\mathcal{L}_{CE}\left(\varphi\right) = -\sum_{i=1, x_i \in \mathcal{Q}}^{K_Q} y_i \log\left(p\left(\hat{y}_i = y_i | x_i, \varphi\right)\right) \tag{12}$$

The word embedding fitting loss $\mathcal{L}_{dissim}\left(\psi\right)$ is defined as dissimilarity between the predicted semantic features and the ground-truth semantic features.

$$\mathcal{L}_{dissim}\left(\psi\right) = \sum_{c=1}^{K}\sum_{i=1}^{N_S} \left(1 - \frac{\widetilde{w_{c_i}} \cdot w_c^T}{||\widetilde{w_{c_i}}|| \cdot ||w_c||}\right) \tag{13}$$

where $\widetilde{w_{c_i}}$ denotes the predicted semantic feature of the **i**-th sample belonging to class $c$, and $w_c$ denotes the ground truth semantic feature of class c. We train the semantic feature learner within each task to produce task-specific parameters. Parameters of semantic feature learner are optimized by minimizing $\mathcal{L}_{dissim}\left(\psi\right)$. Finally, we combine the cross-entropy classification loss and the word embedding fitting loss via weighted sum, $\lambda$=0.5, and train across all tasks to minimize the total loss $\mathcal{L}_{tol}$.

$$\mathcal{L}_{tol} = \lambda\mathcal{L}_{CE} + (1 - \lambda)\mathcal{L}_{dissim} \tag{14}$$

Table 5: The mean accuracy of the joint training method (of the semantic and visual feature learners) on 600 novel episodes with a 95% confidence interval on miniImageNet and CIFAR-FS datasets.

| Model | miniImageNet 5-way | | CIFAR-FS 5-way | |
|---|---|---|---|---|
| | 1-shot | 5-shot | 1-shot | 5-shot |
| TasNet (concatenation) | **78.68 ± 0.59** | **85.89 ± 0.47** | **80.80 ± 0.70** | 84.86 ± 0.60 |
| TasNet(fusion) | 78.24 ± 0.64 | 85.82 ± 0.49 | 80.14 ± 0.72 | 84.46 ± 0.58 |
| Train jointly | 63.48 ± 0.80 | 76.75 ± 0.61 | 64.72 ± 0.77 | 77.70 ± 0.57 |

The experiment results show that the jointly optimization method of the visual and semantic feature learners can not achieve good performance.

