# OpenReview forum: "Few-Shot Classification with Task-Adaptive Semantic Feature Learning"
_ICLR.cc/2022/Conference — ICLR 2022 Submitted_

### Official Review · Reviewer_QNok · 2021-10-26

**Correctness:** 2
**Technical Novelty And Significance:** 2
**Empirical Novelty And Significance:** 2
**Recommendation:** 5
**Confidence:** 3

**Main Review:**

Strengths
- Using additional semantic information such as class label embeddings is interesting in few-shot learning problems.

Weaknesses
- The proposed method seems incremental to me because the only difference from the existing method is that it predicts the semantic features for the query set from the support set and uses it for the few-shot classification.
- There is room for improvement in the clarity of the paper. For example, I didn't know what "semantic features" meant until I read section 3.2.
I recommend describing concrete examples of it at the early stage of this paper such as Introduction, which also clarifies the problem setting tackled in this paper.
In addition, the explanation of the proposed method (Section 3, 3-1, and 3-2) seems unnecessarily redundant.
- The experiments are unconvincing. The effect of using pre-trained feature extractor $E_{\theta}$ is unclear. Did comparison methods use this pre-trained extractor in the experiments? Although the proposed method seems to work well in Tables 1 and 2, I don't know if the semantic features for query samples introduced in the proposed method are working because the description of the comparison methods is insufficient. Did you confirm that the effect of learning the semantics feature learner locally per each task?

Minor Comments
- index $c_i$ is a bit confusing in eq. (2) since $c_i \in$ { $1,\dots, N$ } . ${\cal S} _c$ is undefined in eq. (3).



**Summary Of The Paper:**

This paper proposes a few-shot learning method that incorporates semantic features (i.e., class label embeddings) for both support and query samples.
Although some existing methods use the semantic features for support samples to improve the performance, no methods use them for query samples.
To this end, the proposed method regresses the semantic features for the query samples from the support samples.
Experiments show the effectiveness of the proposed method.

**Summary Of The Review:**

I have some concerns described in Main Review. Therefore, I recommended "3: reject, not good enough".

----After rebuttal----

Thanks for the response. I still have some doubts about the technical novelty of the proposed method, but
given the other reviews and response, and the strength of the experimental results, I increase my score from 3 to 5.

---

> ### Author Response · Authors · 2021-11-20
> **additional experiments and corrections**
>
> $\underline{Response: }$
>
> We appreciate the effort the reviewer put in. And we thank the reviewer for the summary and suggestions, which helps us to further improve our manuscript. Upon the reviewer’s comments, we have made corrections in the revised manuscript. A detailed point by point response is provided (see below). We hope our response has fully addressed the comments from reviewer.
>
> 1. The proposed method seems incremental to me because the only difference from the existing method is that it predicts the semantic features for the query set from the support set and uses it for the few-shot classification.
>
> $\underline{Answer: }$ To alleviate the reviewer’s concern, we have increased the emphasis that the major contribution of our work is that we proposed a task-adaptive semantic feature learning mechanism to learn semantic features for the query samples, which has never been proposed before. The existing multi-modal methods incorporating additional modality to the support samples, while only visual features are available for the query samples. To achieve significant classification results, both support samples and query samples should have good discrimination. Adding semantic features can help generating discriminative features for the query samples, especially when the images of different categories have similar background or irrelevant objects, the efficacy is particularly significant. This is also the key contribution that separates our work from existing multi-modality methods.
>
> We have made some corrections in the introduction section, and supplemented some experiments to further prove the effectiveness of our TasNet in learning discriminative features for query samples with hard tasks (samples with similar background and interference).  The detailed implementation and experiment results is shown in appendix in the revised manuscript. Figure 5 shows the classification result of a hard task, and Figure 6 is the TSNE visualization of the query samples from the hard task.
>
> 2. meaning of "semantic features".
>
> $\underline{Answer: }$ we apologize for improper writing leading to confusion to the reviewer. We have revised the introduction section upon the reviewer’s request to avoid confusion.
>
> 3. explanation of the proposed method
>
> $\underline{Answer: }$ Thanks for pointing this out. In section 3.1 we present some definitions and notations to explain what is the few shot learning, and how the few shot learning model is trained. We believe that some details are worth keeping as we think some details are not universally known by the community. we have made some correction for section 3.1. And section 3.2 presents the overall architecture of our method, we have made some corrections as well.
>
> 4. The effect of using pre-trained feature extractor Eθ.
>
> $\underline{Answer: }$ In order to alleviate the reviewer’s confusion, we have shown the experiment results using the pre-trained extractor as follow. The output of the feature extractor $E_\theta$ is a 640-dimensional feature vector.
>
> Model|	miniImageNet 1-shot|5-shot|	CIFAR-FS 1-shot| 5-shot
> - | :-: | :-:| :-: | :-: |
> Proto+Tasnet (concatenatin)	|78.68 ± 0.59	|85.89 ± 0.47|	80.80 ± 0.70|	84.86 ± 0.60
> Proto+Tasnet (fusion)	|78.24 ± 0.64|	85.82 ± 0.49|	80.14 ± 0.72|	84.46 ± 0.58
> feature extractor $E_\theta$|	54.07 ± 0.79|	74.96 ± 0.61|	62.89 ± 0.93|	78.75 ± 0.65
>
>
> Model	|FC100 1-shot| 5-shot|	CUB 1-shot|5-shot
> - | :-: | :-:| :-: | :-: |
> Proto+Tasnet (concatenatin)|	61.60±0.83	|68.51 ± 0.69|	79.89 ± 0.74|	85.76 ± 0.63
> Proto+Tasnet (fusion)|	60.98 ± 0.89	|67.66 ± 0.84|	79.17 ± 0.67|	84.99 ± 0.66
> feature extractor $E_\theta$	|37.36 ± 0.65	|53.06 ± 0.73|	51.74 ± 0.85	|72.12 ± 0.67
>
> 5. is the semantic features for query samples introduced in the proposed method working.
>
> $\underline{Answer: }$ Regarding to the impact of the semantic feature learner on the overall performance improvement, we have conducted some ablation study in section 4.3.1 and section 4.3.2. To be obviously, we put the results in table.
>
> visual	| semantic	| fused| 	concatenated| 	miniImageNet  1-shot	| 5-shot	|CIFAR-FS  1-shot| 	5-shot
> - | :-: | :-:| :-: | :-: | :-: | :-: | :-: |
> &check;	| | | 	| 		60.37	| 78.02| 72.20| 	83.50|
>  |&check;| | 	| 		75.57| 	83.23| 79.97| 	84.13|
> &check;	| &check;| &check;| | 78.24| 85.82	| 80.14| 84.46|
> &check;| &check;	|  	| &check;	| 78.68| 	85.89| 80.80| 84.86|
>
> From the above table, we can seen that, the result has achieved a significant accuracy when only the sematic features are used, which demonstrates that the semantic features for query samples can certainly work. It is the main reason for the performance boosting.
>
> 6.Minor Comments: index c_i is a bit confusing in eq. (2) since c_i∈ { 1,…,N } . S_c is undefined in eq. (3).
>
> $\underline{Answer: }$ To address the reviewer’s confusion, we have made corrections to the index of x and a. Firstly, we extract base features for all samples in one task. The range of index is $1,…,N(K_S+K_Q)$.

---

### Official Review · Reviewer_7a88 · 2021-10-29

**Correctness:** 3
**Technical Novelty And Significance:** 2
**Empirical Novelty And Significance:** 2
**Recommendation:** 5
**Confidence:** 3

**Main Review:**

Strengths:
+ A task-adaptive semantic feature learner is proposed to incorporate semantic features for both support and query samples.
+ Two modality combination implementations, feature concatenation and feature fusion, are proposed.
+ The visual and semantic features are learned separately, thus preserving the structural heterogeneity of different modalities.
+ Experimental results are provided on four benchmark datasets that demonstrate that the proposed approach outperforms existing approaches.

Weaknesses:
- A clear definition of N-way K-shot few-shot tasks is not provided.
- What does interference mean in the context of Figure 1?
- What is the definition of semantic features? Are they text-based features?
- How are visual-feature class centers and label-derived semantic features computed?
- What is the difference between D_b, D_v, and D_n datasets?
- Effort has to be put in Section 3 to improve its clarity and readability. It is highly recommended to clearly define the problem to be addressed and the approach to be outlined along with any notation in the beginning of the section.
- No discussion of the baselines and their settings is provided. Also, what is the meaning of dashes in Tables 1 and 2?
- It is not clear why jointly learning visual and semantic features will not improve performance?

Some minor comments follow:
- Abstract: "to incorporates" -> "to incorporate"
- pg. 1: Please define acronyms before they are first used, e.g., FSL
- pg. 2: "Few-shot classification methods exhibits" -> "Few-shot classification methods exhibit"
            "learning a discriminative representations" -> "learning discriminative representations"
- pg. 3: "the these methods is" -> "these methods is"
- Table 2 caption: "in Tabel1" -> "in Table 1"
- pg. 9: "except" -> "expect"
- pg. 6: "We computes" -> "We compute"

**Summary Of The Paper:**

This paper proposes a task-adaptive semantic feature learning mechanism to incorporate semantic features for both support and query samples. Two modality combination implementations, feature concatenation and feature fusion, are devised based on the semantic feature learner. Experimental results are provided on four benchmark datasets that demonstrate that the proposed mechanism outperforms existing methods.

**Summary Of The Review:**

Overall, I recommend the rejection of the paper. My major concern is that the main contribution of the paper, i.e., separately learning the visual and semantic features, is not very intuitive and well justified. Furthermore, the proposed modality combination approaches are well-known methods.

----After rebuttal----

I would like to thank the authors for their effort to answer my questions. Even though the technical novelty of the paper is limited, as also supported by the new experiments added, the responses to my questions are satisfying. Thus, I increase my score from 3 to 5.

---

> ### Author Response · Authors · 2021-11-19
> **additional experiments and corrections**
>
> $\underline{Response: }$
>
> We thank the reviewer for the comments. We acknowledge the defects and problems in our paper. To address the comments, we have made some corrections in the revised manuscript. A detailed point by point response is provided. We hope our response has fully addressed the comments from reviewer.
>
> 1. A clear definition of N-way K-shot few-shot tasks is not provided. What is the difference between D_b, D_v, and D_n datasets?
>
> $\underline{Answer:  }$ we apologize for our improper writing that result in the confusion to the reviewer. We have rewriting the section 3.1 problem definition to clarify these problems.
>
> 2. What does interference mean in the context of Figure 1?
>
> $\underline{Answer:  }$ We apologize for this confusion, and we have added explanation about it below Figure 1 in the revised version. Interference refers to objects in the image that are irrelevant to the image category. In the right part of Figure 1, there are five different categories. However, they all have flower in the images, which will mislead the model to recognize these samples as flower categories or classify them into one class, i.e. flower is the interference for these samples.
>
> 3. What is the definition of semantic features? Are they text-based features?
>
> $\underline{Answer:  }$ We acknowledge that our definition of semantic features is not very clear. To address this, we have added the following in Section 3.3: "Semantic features is the word embedding of the text label. It is predicted by the Glove embedding method."
>
> 4. How are visual-feature class centers and label-derived semantic features computed?
>
> $\underline{Answer:  }$ We apologize for this confusion. Visual-feature class centers are computed via mean computation. It is show in Eq (3). The label-derived semantic features are predicted via $g_\psi$ network. The detailed implementation is show in section 3.2 and Figure 2. The semantic feature learner $g_\psi$ is firstly trained with the support samples, show in $F(\cdot|\mathcal{S})$ module. Then the trained network is used to predict the semantic features for the query samples. We have made corrections and highlighted them in the revised manuscript.
>
> 5. Effort has to be put in Section 3 to improve its clarity and readability. It is highly recommended to clearly define the problem to be addressed and the approach to be outlined along with any notation in the beginning of the section.
>
> $\underline{Answer:  }$ Thanks for your suggestion. Actually, we have stated the problem and our method at the beginning of section 3. The problem that we want to solve is improving the feature discrimination of the query samples by training a task-adaptive semantic feature learner which is used to predict semantic features for the query samples.
>
> Our model trains the visual feature learner across all tasks, and trains the semantic feature learner within each task. At each episode, the semantic feature learner is trained with support samples and label’s word embeddings. Then the trained network is used to predict the semantic features for the query samples.
>
> We apologize for missing some necessary definitions and notations at this section. In the revised manuscript, we have rewritten the section 3.1 to clarify these problems. Detailed modification can be seen in the answer to question 1.
>
> 6. No discussion of the baselines and their settings is provided. Also, what is the meaning of dashes in Tables 1 and 2?
>
> $\underline{Answer:  }$  Thanks for pointing this out. Our method is implemented with ProtoNet as the baseline. The backbone of ProtoNet is ResNet-12, and the output feature is 640-dimensional. Dashes in Tables 1 and 2 (“-"\ ) indicates that these models have not run experiments on these datasets.
>
> 7. It is not clear why jointly learning visual and semantic features will not improve performance?
>
> $\underline{Answer:  }$ In order to alleviate the reviewer's concern, we have made some experiments to jointly train the visual and semantic features learner. The detailed implementation of the method and the experiment results are presented in the appendix of the revised manuscript.
>
> Model	|miniImageNet 1-shot| 5-shot  |    CIFAR-FS 1-shot|  5-shot
> - | :-: | :-:| :-: | :-: |
> Train jointly|	63.48 ± 0.80|	76.75 ± 0.61|	64.72 ± 0.77	|77.70 ± 0.57
>
> The performance is rather poor.
>
> 8. Some minor comments follow
>
> $\underline{Answer:  }$ We would like to thank the reviewer for the careful observation, which helps us correct the mistakes in the manuscript. We have made corrections accordingly in the revised manuscript.
>
> A major contribution of our work is that we proposed a task-adaptive semantic feature learning mechanism to learn semantic features for the query samples, which has never been proposed before. To achieve significant classification results, both support samples and query samples should have good discrimination. Adding semantic features can help generating more discriminative features for the query samples.

---

### Official Review · Reviewer_LGLG · 2021-11-02

**Correctness:** 4
**Technical Novelty And Significance:** 2
**Empirical Novelty And Significance:** 2
**Recommendation:** 6
**Confidence:** 4

**Main Review:**

The paper proposed a reasonable model to use the text labels in image classification. The experimental results look convincing. With that said, the proposed solution looks overly complicated.  A simple baseline should be considered: a multi-task learning model with two outputs, one to predict the image class,  the other to fit the Glove embedding. Compared with this simple baseline, the extra complicity of the proposed model should be better motivated. Also it would be good to have an empirical comparison with the simple two-outputs baseline.



**Summary Of The Paper:**

In few-shot learning for image classification, visual features alone may represent multiple objects in an image. The label text provides additional information that could serve as useful inductive bias, indicating which object in an image the model should pay attention to. There are multiple ways to use such text information to help few-shot classification. The authors propose to build a model that predicts both the image classes and the text label (or its Glove embedding). In other words, an additional loss term is introduced to regulate the learning of visual features. The authors show that the additional "semantic features" (because Glove embedding is used) help improve few-shot image classification.

**Summary Of The Review:**

The authors use additional information in the training labels to improve few-shot image classification. The solution is a combination of different existing tricks. Without proper ablation experiments, it is difficult to infer generalisable knowledge from the results.

---

> ### Author Response · Authors · 2021-11-19
> **additional experiments and corrections**
>
> $\underline{Response: }$
>
> We greatly appreciate the reviewer’s constructive recommendations. And we thank the reviewer for the summary. A detailed point by point response is provided (see below). We hope our response has fully addressed the comments from reviewer.
>
> 1. A simple baseline should be considered: a multi-task learning model with two outputs, one to predict the image class, the other to fit the Glove embedding. Compared with this simple baseline, the extra complicity of the proposed model should be better motivated. Also it would be good to have an empirical comparison with the simple two-outputs baseline.
>
> $\underline{Answer: }$ We thank the reviewer for the suggestion. the multi-task learning model, from my understand, is a model that train the semantic feature learner and the visual feature learner jointly. Actually, we have tried the multi-task learning method before. However, the experiment results are not ideal, hence we abandoned this idea. The detailed implementation of our multi-task learning method is as follow.
>
> We constructed a cross entropy loss to optimize the visual feature learner, and a word embedding fitting loss, i.e. the cosine dissimilarity loss to optimize the semantic feature learner.
>
> The loss $\mathcal{L}_{CE}$ for training visual feature learner is defined as a cross-entropy classification loss averaged across all query-support pairs. It compares the Euclidean distance between the query features and the class mean of the support features to get the predicted score.
>
> The word embedding fitting loss $\mathcal{L}_{dissim}\left(\psi\right)$ is defined as dissimilarity between the predicted semantic features and the ground-truth semantic features.
>
> Finally, we combine the cross-entropy classification loss and the word embedding fitting loss via weighted sum, $\lambda$=0.5, and train across all tasks to minimize the total loss $\mathcal{L}_{tol}$.
>
> $ (1-\lambda) \mathcal{L}_{dissim} + \lambda\mathcal{L}_{CE}$
>
>
> Model	|miniImageNet 1-shot|	miniImageNet 5-shot  |    CIFAR-FS 1-shot| CIFAR-FS 5-shot
> - | :-: | :-:| :-: | :-: |
> Proto+Tasnet (concatenatin) |	78.68 ± 0.59|	85.89 ± 0.47|	80.80 ± 0.70|	84.86 ± 0.60
> Proto+Tasnet (fusion)	|78.24 ± 0.64|	85.82 ± 0.49	|80.14 ± 0.72|	84.46 ± 0.58
> Train jointly|	63.48 ± 0.80|	76.75 ± 0.61|	64.72 ± 0.77	|77.70 ± 0.57
>
> The experiment results show that jointly optimize the visual and semantic feature learner can not achieve good performance.
>
> As for the comparison with the simple two-outputs baseline, we have made some ablation studies to compare the impact of each modality on the overall performance quantitatively and qualitatively. Detailed analysis is shown in section 4.3.1 EFFECTIVENESS OF THE SEMANTIC FEATURE LEARNER, and section 4.3.2 T-SNE VISUALIZATION. We compare four kinds of features: only the visual feature, only the semantic feature, fused feature of the semantic and visual features, concatenated feature of the semantic and visual features, to show the performance they can achieve.
>
> From the quantitative and qualitative results, we can conclude that the task-adaptive semantic feature learning mechanism is the main reason for the performance boosting.
>
> 2. The authors use additional information in the training labels to improve few-shot image classification. The solution is a combination of different existing tricks. Without proper ablation experiments, it is difficult to infer generalizable knowledge from the results.
>
> $\underline{Answer: }$  We apologize for our improper writing causing misunderstanding to the reviewer. Indeed, a major contribution of our work is that we proposed a task-adaptive semantic feature learning mechanism to learn semantic features for the query samples, which has never been proposed before. To achieve significant classification results, both support samples and query samples should have good discrimination. Adding semantic features can help generating more discriminative features for the query samples, especially when the images of different categories have similar background or similar irrelevant objects, the efficacy is particularly significant compare with the previous methods. This is also the key contribution that separates our work from existing multi-modality methods. We have made some corrections in the introduction section.
>
> Regarding to the ablation experiments, we have made some ablation studies in section 4.3.1 and section 4.3.2, actually. To alleviate the reviewer’s concern, we have supplemented an additional experiment conducted on hard tasks, which is constructed of samples with similar background and interference. Since pictures can not be shown here, The detailed implementation and experiment results is shown in appendix in the revised manuscript. Figure 5 shows the classification result of a hard task, and Figure 6 is the TSNE visualization of the query samples from the hard task.

---

### Official Review · Reviewer_wrVH · 2021-11-03

**Correctness:** 3
**Technical Novelty And Significance:** 2
**Empirical Novelty And Significance:** 3
**Recommendation:** 6
**Confidence:** 4

**Main Review:**

Strength:
The paper is clearly written and easy to follow.
Empirically they show that using such semantic features clearly shows improvements over the visual features.

Questions:
(i) The method (TASNet) is built over Protonet (Snell et al) method. Does it show similar performance gain with other popular architectures such as MetaOptNet, etc.?
(ii) Details on the architectures of visual feature learner (f) and semantic feature learner (g) are missing?

Weakness:
The motivation for the paper involves robust few-shot learning with different categories under similar background, interference. The results are only on the full test set. Results on a few episodes where such conditions exist would cement the robustness of the method.

**Summary Of The Paper:**

This paper describes a few-shot learning approach that takes into account the semantic relatedness of the ground truth labels rather than just using them as binary labels. In the few-shot learning setting, the paper describes a method to train the semantic feature learner (on labels) during the meta-training phase and then at meta-test time use the semantic feature learner to get the said features for the query images. This way both the support and query examples would have semantic features and empirically it has shown to work on many few-shot benchmarks on computer vision datasets.

**Summary Of The Review:**

Overall the paper suggests a method that makes an improvement over an existing state-of-the-art, but the efficacy of their enhancements are not fully tested. In principal adding textual data that is semantically less ambiguous than image data helps.

---

> ### Author Response · Authors · 2021-11-19
> **Additional experiments**
>
>
>
> $\underline{ Response:}$
>
> We would like to thank the reviewer for providing valuable critiques and suggestions, which further strengthen the revised manuscript. We have updated our manuscript based on these suggestions. A detailed point by point response is provided (see below). We hope our response has fully addressed the comments from reviewer.
>
> 1.  The method (TASNet) is built over Protonet (Snell et al) method. Does it show similar performance gain with other popular architectures such as MetaOptNet, etc.?
>
> $\underline{Answer:  }$ we are grateful for the reviewer's suggestions. We have adopted the new experiment on MetaOptNet and TADAM architectures to further prove the effectiveness and generalization of TasNet.
>
> Model | miniImageNet 1shot | miniImageNet 5shot | CIFAR-FS 1shot | CIFAR-FS 5shot
> - | :-: | :-:| :-: | :-: |
> TADAM +Tasnet | 78.57 ± 0.56 |  85.57 ± 0.41| 80.47 ± 0.67 | 84.57 ± 0.59
> MetaOptNet +Tasnet | 78.36 ± 0.61 |  85.78 ± 0.52| 80.76 ± 0.73 | 84.69 ± 0.63
>
> Model | FC100 1shot | FC100 5shot | CUB 1shot | CUB5shot
> - | :-: | :-:| :-: | :-: |
> TADAM +Tasnet | 61.79 ± 0.78|68.35 ± 0.65|79.72 ± 0.59|86.12 ± 0.68
> MetaOptNet +Tasnet | 61.24 ± 0.91|67.98 ± 0.73|79.32 ± 0.71|84.75 ± 0.72
>
> TADAM +Tasnet and MetaOptNet +Tasnet are the experiments based on the feature concatenation of different modality with MetaOptNet and TADAM as the baseline methods. The results proves that the TasNet method is generalizable to some baseline methods.
>
> 2.  Details on the architectures of visual feature learner $f_\varphi$ and semantic feature learner $g_\psi$ are missing?
>
> $\underline{Answer:  }$ We apologize for missing details on the architectures of visual feature learner $f_\varphi$ and semantic feature learner $g_\psi$ in the previous manuscript. We have supplemented a subsection in section 4, titled 4.1 architectures, in the revised manuscript, to describe detailed architectures of all networks mentioned in our model.
>
> 3.The motivation for the paper involves robust few-shot learning with different categories under similar background, interference. The results are only on the full test set. Results on a few episodes where such conditions exist would cement the robustness of the method.
>
> $\underline{Answer:  }$ We thank the reviewer for the good suggestion. We have supplemented some experiments upon the reviewer’s request to cement the robustness our method. Firstly, we have constructed a sub dataset (we name it hard dataset) from the test split of miniImagenet dataset manually. It contains 8 classes, including lion, golden retriever, goose, coral reef, vase, spider web, ant, house finch. Among samples from all these categories, some of samples from two or more categories have similar background or similar interference, for example, vase and ant categories, flowers are the common interference in the images from these two classes, and for the bird and spider web categories, trees and grassland are common similar backgrounds in the images from these two classes. Then we construct 20 tasks from this hard dataset, and compare our TasNet with ProtoNet and AM3 methods on this dataset. AM3 is a method that combine semantic features for the support samples, while only visual features are extracted from the query samples. The mean accuracy is show in table 3.
>
> |Model	|Hard task 1-shot| Hard task 5-shot|
> - | :-: | :-:|
> |ProtoNet	|44.93 ± 0.71	|70.67 ± 0.58
> |ProtoNet+AM3	|48.11 ± 0.69	|71.24 ± 0.61
> |ProtoNet+Tasnet|	50.97 ± 0.76	|72.02 ± 0.69
>
> To further validate the effectiveness of TasNet in discriminating different targets from similar background and interference, we have conducted an additional experiment. In this experiment, we show the classification results of one of the hard tasks in Figure 5. We carried the comparison experiments on 5-way 1-shot case. Five categories of ant, house finch, spider web, vase, coral reef, are included in this task. Each class has 1 support samples and 15 query samples, that is the number of test samples is 5$\times$15=75.
>
> Since pictures can not be shown here, we have put  Figure 5 and the TSNE visualization of the query samples from hard task is shown in  Figure 6 in appendix of the revised manuscript.
>
> In section 4.4.1 and section 4.4.2, we have made some ablation studies, which shows that the task adaptive semantic feature learner is certainly having the potential to identify the objects in the images.
>
> 4. Overall the paper suggests a method that makes an improvement over an existing state-of-the-art, but the efficacy of their enhancements are not fully tested. In principal adding textual data that is semantically less ambiguous than image data helps
>
> $\underline{Answer:  }$To address the reviewer’s concern, we have further proved the efficacy of TasNet in hard task which is constructed of samples with similar background and interference. Detailed experiment results are shown above. We have updated all the experiment result in the revised manuscript.

---

### Decision · Program_Chairs · 2022-01-20

**Decision:**

Reject

**Comment:**

This paper proposes a few-shot learning method that learns task-adaptive semantic features that can incorporate for both of the support and query sets. Two approaches for modality combination are developed. The additional experiments in the author response addressed some concerns of the reviewers. However, the technical novelty of the proposed method is high enough since the proposed method uses existing techniques.